# Spatiotemporal Patterns of Menin Localization in Developing Murine Brain: Co-Expression with the Elements of Cholinergic Synaptic Machinery

**DOI:** 10.3390/cells10051215

**Published:** 2021-05-16

**Authors:** Shadab Batool, Jawwad Zaidi, Basma Akhter, Anosha Kiran Ulfat, Frank Visser, Naweed I. Syed

**Affiliations:** 1Hotchkiss Brain Institute, University of Calgary, Calgary, AB T2N 4N1, Canada; shadab.batool@ucalgary.ca (S.B.); jawwad.zaidi@ucalgary.ca (J.Z.); Basma.akhter@ucalgary.ca (B.A.); akulfat@ucalgary.ca (A.K.U.); fvisser@ucalgary.ca (F.V.); 2Department of Neuroscience, University of Calgary, Calgary, AB T2N 4N1, Canada; 3Department of Cell Biology and Anatomy, University of Calgary, Calgary, AB T2N 4N1, Canada

**Keywords:** development, menin, cholinergic synaptic machinery

## Abstract

Menin, a product of *MEN1* (multiple endocrine neoplasia type 1) gene is an important regulator of tissue development and maintenance; its perturbation results in multiple tumors—primarily of the endocrine tissue. Despite its abundance in the developing central nervous system (CNS), our understanding of menin’s role remains limited. Recently, we discovered menin to play an important role in cholinergic synaptogenesis in the CNS, whereas others have shown its involvement in learning, memory, depression and apoptosis. For menin to play these important roles in the CNS, its expression patterns must be corroborated with other components of the synaptic machinery imbedded in the learning and memory centers; this, however, remains to be established. Here, we report on the spatio-temporal expression patterns of menin, which we found to exhibit dynamic distribution in the murine brain from early development, postnatal period to a fully-grown adult mouse brain. We demonstrate here that menin expression is initially widespread in the brain during early embryonic stages, albeit with lower intensity, as determined by immunohistochemistry and gene expression. With the progression of development, however, menin expression became highly localized to learning, memory and cognition centers in the CNS. In addition to menin expression patterns throughout development, we provide the first direct evidence for its co-expression with nicotinic acetylcholine, glutamate and GABA (gamma aminobutyric acid) receptors—concomitant with the expression of both postsynaptic (postsynaptic density protein PSD-95) and presynaptic (synaptotagamin) proteins. This study is thus the first to provide detailed analysis of spatio-temporal patterns of menin expression from initial CNS development to adulthood. When taken together with previously published studies, our data underscore menin’s importance in the cholinergic neuronal network assembly underlying learning, memory and cognition.

## 1. Introduction

It is becoming increasingly evident that genes that were initially thought to play specific and restricted roles in any given tissue may in fact serve myriad functions at various stages of an animal’s life span. For instance, multiple endocrine neoplasia type 1 (*MEN1*)—a tumor suppressor gene, which is mutated in patients with *MEN1* syndrome [1]—is expressed in almost all developing tissues and organs [2]. *MEN1* knockout generates a lethal phenotype, thus making it difficult to decipher its precise function in the intact animal [3]. Moreover, *MEN1* orthologues have been found in *Drosophila* [4], *Lymnaea* [5], rodents [6], etc., demonstrating that functionally this gene is evolutionarily highly conserved. Recent investigations into menin’s crystal structure and its physiological aspect of crosstalk in myriad signaling pathways [7] have shed some further light onto its role in gene expression and cell–cell signaling involved in various important cellular pathways [8]. It is interesting to note that although menin expression in the brain is the highest during development, it does nevertheless become restricted to only cortical and hippocampal regions in the adult CNS, suggesting a potential role in regulation of neurite growth, synapse formation, synaptic plasticity [9], learning, memory and cognition [3] in a host of animal models. Consistent with this postulate, recent studies [10] have identified a homologue of *MEN1* gene in the mollusc *Lymnaea* (L-menin), which was differentially expressed in postsynaptic neurons exhibiting cholinergic phenotype. This study provided the first direct evidence that *MEN1* gene is indeed expressed and upregulated in *Lymnaea* neurons in vitro during cholinergic synaptogenesis [11]. Later, this group demonstrated that the trophic factor and activity-dependent mechanisms underlie menin function regulation, which was required for the expression of nAChR receptors in both invertebrates [5] and vertebrates [6]. Menin was subsequently shown to be cleaved by Calpain into C (C-terminal) and an N (N- terminal) fragments; the former was demonstrated to target nAChRs to postsynaptic sites in the hippocampus [6]. Similarly, Xu et al. demonstrated menin’s involvement in plasticity and neuropathic pain in spinal cord neurons following peripheral nerve injury in a rodent model [9].

Menin was also found to control the transcriptional regulation of nAChR α5 subunit and the physiological clustering of α7 subunit of nAChR at the glutamatergic presynaptic terminals in mouse hippocampal cultures [6], though its spatio-temporal association with synaptic proteins remained unknown. A recent study from another group has since shown that *MEN1* conditional neuronal knockout in a mouse brain leads to significant decrease in dendritic branching and spine formation thus impacting neuronal growth, synapse function, and plasticity [3]. Taken together, these studies demonstrate menin’s diverse roles in the CNS specifically during development, synaptic maturation and functions. However, neither of these studies have specifically demonstrated the spatio-temporal patterns of menin expression in a developing brain, nor have they corroborated its expression with other synaptic proteins such as SYT-1 (Synaptotagmin), postsynaptic density protein 95 (PSD-95) and nAChR, which are all essential players in synapse formation and synaptic plasticity underlying learning and memory.

Since the C-menin fragment appears to be targeted to specific cholinergic synaptic sites in the adult brain, its precise expression patterns in conjunction with other components of the synaptic machinery in the developing brain are also yet to be demonstrated. In the absence of this information, the precise role of menin, particularly the C-fragment in synapse formation, synaptic plasticity and learning and memory remains thus incomplete. This study was designed to decipher the spatio-temporal patterns of menin during various brain developmental stages in a mouse model. We specifically sought to define the C-fragment expression patterns at various developmental stages when neither are neurons fully differentiated, nor targeted to distinct brain regions, or have yet established synaptic connectivity? Further, we aimed to deduce whether menin expression coincides in concomitant with the expression of other pre-and postsynaptic proteins and nAChR at excitatory and or inhibitory neurons in the CNS.

In this study, we provide the first direct evidence vis-à-vis the developmental expression of C-menin along with other synaptic proteins in a mouse brain. We also provide the first direct evidence that during various developmental stages, even in an undifferentiated brain, menin is often co-localized with other synaptic proteins at both premature and matured synapses in the hippocampus, cortex, cerebellum and thalamus—predominantly involving the nAChR that are thought to be involved in learning, memory and cognition. This study thus provides novel insights into a tumor suppressor which likely plays myriad roles not just in a developing organism but also serves essential functions in the adult nervous system.

## 2. Materials and Methods

### 2.1. Animals: Brain Slices and Neuronal Cell Culture

All protocols, experiments, animal use and care were followed as per the University of Calgary and Canadian Council on Animal Care animal care guidelines. Different ages of mice, ranging from embryonic day 12.5 (E12.5), E15.5, E18, postnatal day 10 (P10), to an adult C57BL/6 (Charles River, Calgary, AB, Canada) were used for experiments involving brain slices. Embryos were dissected from the pregnant female mice which were anesthetized with isoflurane and sacrificed by decapitation to harvest the brain tissue. The brains obtained were immediately transferred to vials with 4% paraformaldehyde (PFA) for an overnight fixation at 4 °C and then kept in 2% sucrose solution submerged. These fixed brains were then snap-frozen using dry ice and OCT compound block and stored at −80 °C till required for brain sectioning. Brain slices, 16 µm were prepared at −20 °C using cryostat as previously described [12].

Embryos were dissected from C57BL/6 (Charles River) pregnant females on E18 for hippocampal neuronal cultures. Pregnant mice were anesthetized with isoflurane and sacrificed by decapitation to harvest embryos brains. E18 embryos were immediately dissected and sacrificed by decapitation and hippocampi isolated. Hippocampal tissue was dissected in solution (1 × HBSS containing 10 mM HEPES; 310 mOsm, pH 7.2), and enzyme treated with papain (50 U/mL), 150 mM CaCl_2_, 100 µM L-cysteine, and 500 µM EDTA in neurobasal medium (NBM) for 20 m at 37 °C (incubator). The tissue was then washed 3 times with NBM supplemented with 4% FBS, 2% B27, 1% penicillin-streptomycin, and 1% L-glutamine (GIBCO). Neurons were dissociated by trituration with fire-polished glass Pasteur pipettes and plated at around 900 cells/mm^2^ for achieving lower density cultures onto glass coverslips which had been washed previously with nitric acid and coated with poly-D-lysine (30 µg/mL; Sigma Aldrich, Ontario, Canada) and laminin (2 µg/mL; Sigma Aldrich, Ontario, Canada) in costar 12-well plates (VWR) in NBM supplemented as aforementioned. The culture media was changed to NBM supplemented with 2% B27, 1% penicillin-streptomycin, and 1% L-glutamine the very next day and neurons were maintained throughout at 37 °C with 5% CO_2_ and ~50% of the media was changed every 3–4 days.

### 2.2. Immunocytochemistry and Immunohistochemistry

Hippocampal neuronal cultures were fixed at Day in vitro (DIV) 3–12 for 30 min. with 4% paraformaldehyde and 0.2% picric acid (Sigma Aldrich) in 1 × PBS, and permeabilized for 1 h with incubation medium (IM) containing 0.5% Triton and 10% goat serum in 1 × PBS. Negative controls were performed to test the specificity of the antibodies as described previously [6]. Primary antibodies (menin C-terminal epitope (Bethyl Laboratories, A300–105A); menin C-terminal epitope (Santa Cruz, CA, USA, SC-374371); α-neurofilament (Novus Biologicals, Centennial, CO, USA, NB300-222); α-synaptotagmin (EMD Millipore, Burlington, MA, USA, MAB5200); α-PSD-95 (Antibodies Incorporated, Davis, CA, USA, 75-028); α-nAChR α5 (AbCam, Burlington, MA, USA, ab41173-100); α-GluR1(Abcam, MA, USA, ab183797), α-mGluR1(Abcam, MA, USA, ab82211); α-GABAA (Abcam, Burlington, MA, USA, ab-252430); α-GABA B(Abcam, MA, USA, ab55051)) were used at 1:500 in IM for 1 h. Secondary antibodies (Alexa Fluor 488, 568, or 680 conjugated goat α-rabbit, α-mouse or α-chicken (Invitrogen, Vancouver, BC, Canada) were used at 1:100 in IM for 1 h. α7-nAChR were labeled with Alexa Fluor 555 conjugated α-Bungarotoxin (Invitrogen, BC, Canada, B35451) at 2 µg/mL in IM for 1 h. Neuronal cultures were then subjected to three 15 m washes in 1 × PBS after each incubation at room temperature. Cells were mounted using ProLong Gold antifade reagent with DAPI (Invitrogen, BC, Canada).

For brain slices, sections were exposed to freshly made 0.3% hydrogen peroxide in 0.1% sodiumazide for 15 m to block any endogenous peroxidase activity to avoid background labeling of blood vessels. Slices were then subjected to heat mediated antigen retrieval step in sodium citrate buffer for 10 min. and washed with 1 × PBS for 15 min. The rest of the protocol for labeling tissue was same as mentioned above.

The specificity of the antibodies used in this study has previously been confirmed using Western blot analysis [6]. All antibodies were, however, further optimized on sagittal and coronal 16 µm thick, adult mouse brain slices in the AP Research Lab of Alberta Precision Labs (Calgary, AB, Canada) using high precision, reliable and automatic methods as described above.

### 2.3. Quantitative PCR (qPCR)

Total RNA was isolated from E16, E18, P10 and adult C57BL/6 mice brains specifically from hippocampal, cortical, thalamic and cerebellar regions using the RNeasy micro kit (QIAGEN, Ontario, Canada) according to the manufacturer’s recommendations. Reverse transcription was performed using the quantitative reverse transcription kit (QIAGEN, Ontario, Canada). For the negative control groups, all components except the reverse transcriptase were included in the reaction mixtures. qPCR was performed with KAPA SYBRgreen Fast Universal qPCR kit (Kapa Biosystems, Wilmington, MA, USA) and primers directed to a region of 80–120 bases. Human glyceraldehyde 3-phosphate dehydrogenase (GAPDH) gene was used as the reference gene. Control reactions and those containing cDNA from cerebral arteries were performed with 1 ng of template per reaction. The running protocol extended to 45 cycles consisting of 95 °C for 5 s, 55 °C for 10 s, and 72 °C for 8 s using an Eppendorf, Hartford, CT, USA Realplex 4 Mastercycler. PCR specificity was checked by dissociation standard curve analysis. Assay validation was confirmed by testing serial dilutions of pooled template cDNAs, suggesting a linear dynamic range of 2.8–0.0028 ng of template. Efficiency values for qPCR primers ranged between 85–110% (R^2^ = 0.97–1.00). Expression of *MEN1* gene in different regions of the brain relative to GAPDH was determined using the relative expression software tool (REST) version 2.0.13 [13].

### 2.4. GFP-Cre_AAV Stereotaxic Injections in Homozygous Floxed MEN1 Mice

The mice underwent stereotaxic injections with adeno-associated virus (AAV) designed to express GFP-Cre under a Synapsin promoter to specifically eliminate MEN1 expression in the hippocampal CA1, CA3 and DG neurons. The AAV capsids efficiently allowed for gene transfer to neurons of interest only in the hippocampal CA3-CA1, DG regions and resulted in neuron-specific genetic deletion. The AAV transfer plasmid pENN.AAV.hSyn.HI.GFP-Cre.WPRE.SV40 (Addgene, Watertown, MA, USA, number 105540) was packaged into AAV serotype 9 with a titer of ≥1 × 10¹³ vg/mL. The AAV was introduced into homozygous floxed MEN1 mice and stereotactically injected bilaterally into the dorsal hippocampi (2 mm behind bregma, 2 mm lateral, and 1.6 mm below the dural surface for CA1, and 1.9, 2.1 and 2.1 mm, respectively, for CA3. The injections were performed with glass micropipettes with the tip diameter ranging between 10 to 20 μm and injected slowly with a pressure-injection system. The recovery time post-injections ranged between 3–5 weeks and the first step to validate this model was to determine if the virus was indeed delivered to the CA1 and CA3 regions followed by determining the expression of menin protein using IHC as described above.

### 2.5. Microscopy

Confocal microscopy [14] was performed at 60X magnification on neuronal cultures and brain slices (16 µm), as previously described [6]. Fluorophores were excited with 402, 488, 568 and 680 laser wavelengths and emissions collected through 450/50, 525/50 and 700/75 filter cubes. Imaging parameters including field of view size, laser intensity and channel gain were kept strictly the same amongst relevant samples. Images were collected from 12 samples prepared from independent culture sessions.

For 16 µm thick brain slices fluorescence, Olympus VS120 high-capacity slide scanner microscope was used to scan different slices DAPI, FITC, TRITC, cy5 fluorescence filters. All images were obtained using same parameters (amongst relevant samples) including the exposure time, defibrillator (to minimize all background noise) and field of view size. Bright field images at 40× were obtained from samples that had undergone chromogen immunohistochemistry. All microscope settings are shown in Table (see Appendix A).

Menin fluorescence intensity was measured and quantified using IMAGE J. The fluorescence range was from 0–250, where AU < 20 was considered background signal, AU <20 and >30 moderate and values >30 were deemed strong throughout the quantification of relevant samples. Brain slices were also imaged using confocal microscopy to measure the degree of colocalization between different proteins. All brain regions at different ages were labeled using mouse brain ATLAS as a reference [15].

### 2.6. Experimental Design and Statistical Analysis

Data sets were derived from ≥8 independent experiments, using samples from ≥12 independent cell culture preparations or tissue collected from ≥13 animals, to ensure that all results were reliable and replicable. Minimum sample sizes for quantitative data were determined using the resource equation method. Image processing, fluorescence intensity and degree of colocalization (Pearson’s co-efficient) using Jacop Plugin as described previously [16] were performed with ImageJ (NIH). Data analyses were performed blinded by acquisition file number. For brain slices, equal area for the region of interest (ROI) was selected randomly (independent of the size of the tissue), *n* ≥ 12 every slice per region to minimize biased. Quantification tools were kept constant for all the tissues amongst relevant samples.

Statistical analyses were performed using SPSS Statistics v22 for Mac and Prism8 version 8.3.1Graph pad software. The data distribution was analyzed with the D’Agostino and Pearson [17] test of normality, Bartlett’s test for homoscedasticity and parametric (*p* > 0.05) or non-parametric (*p* < 0.05) statistical tests were used as appropriate. Differences in fluorescence intensity for ICC were assessed with the one-way ANOVA followed by post-hoc Tukey test. Incidence of colocalization was obtained from Pearson’s co-efficient using JACOB plugin (Bolte S 2006) on IMAGE J, Java 1.8. Significant differences in relative gene expression of MEN1 in different regions of the brain were temporally determined using one-way ANOVA followed by post-hoc Tukey test.

## 3. Results

### 3.1. Specificity of Menin Expression Pattern

A large gap in our understanding of menin, specifically the C-fragment and its involvement in brain development, synapse formation and learning and memory is due to the lack of morphological evidence obtained from various developmental stages in the mammalian brain. Additionally, the spatio-temporal expression patterns of menin protein have not been investigated thoroughly—either in isolation or in conjunction with other players of the synaptic machinery. Here, we first set out to define menin’s expression patterns in mouse brain slices at various developmental stages using immunohistochemistry. To ensure the specificity of the antibody labeling for menin protein, we used a CamkIIa-Cre crossed with MEN1 homozygous mouse to knockdown MEN1 particularly in the hippocampus using Cre AAV (Figure 1Ai,Bi). Low or no staining was observed in MEN1^−/−^ Cre^+/−^ (Figure 1Bi–iii), compared to highly specific expression in Cre negative controls (Figure 1Ai–iii). These data not only demonstrate that MEN1 can be conditionally knockdown in the hippocampus but also validate the specificity of the menin antibody.

### 3.2. Spatiotemporal Specific Localization of MEN1 Gene and Menin Protein Fragments in Neurons

To confirm the specificity of both C and N epitope antibodies further, and their respective cytoplasmic and nuclear localizations, we cultured hippocampal neurons in vitro. On Day In Vitro 7 (DIV 7), neurons were double stained with both N and C menin antibodies. As shown previously by our group, both antibodies specifically labeled their respective fragments (*n* = 18 images, 5 independents samples, Figure 2Ai–iii) in cultures and in intact mouse brain (Figure 2Bi,ii,Ci–iii). Specifically, whereas the N fragment localization was restricted to the nuclear region (Figure 2Ai, the C fragment was distributed throughout the cytoplasmic regions in both the soma and the extrasomal regions (*n* = 18 images, 5 independent samples, Figure 2Aii,iii,Bii. It is also interesting to note that C-menin fragment localization was also evident in little puncta formulations, which are presumed to be synaptic structures (*n* = 18 images, 5 independent samples, Figure 2Aii). To establish this pattern of both C and N menin expression further in the adult brain, these fragments were also labeled in the adult mouse brain using immunohistochemistry (IHC) assay (*n* = 18 images, 5 independent samples, Figure 2 Bi,ii,Ci-iii).The IHC and ICC results revealed that N-terminal epitope of menin localization in the nuclear and perinuclear region of the neurons, whereas the C-terminal menin expression was also observed in the nuclear, perinuclear, cytoplasmic and synaptic regions. (*n* = 18 images, 5 independent samples, see Figure 2A,B,Ci–iii).

To confirm the above morphological data further at the molecular level, we next sought to determine the expression patterns of *MEN1* gene in a spatiotemporal manner. Different aged mouse brain regions were carefully isolated at E16, E18, P10 stages and qPCR analysis was performed (*n* = 6, 3 independent experiments each, triplicate replicates; see Appendix A). The results indicated that indeed *MEN1* gene expression increases periodically in the mouse brain with the highest expression being in an adult brain—specifically in the hippocampus, cortex, thalamus and cerebellum (see Figure 2D). These data suggest that *MEN1* gene expression is likely distinct to these specific brain regions.

### 3.3. Neuron Specific Expression of α-C-terminal Menin in Hippocampal Neurons (In-Situ)

To further confirm whether the menin α-C-terminal signal is localized specifically to neurites, we performed IHC in mouse hippocampal slices and co-labeled it with neurofilament (neuronal marker). Our results from confocal high-resolution images exhibited significant colocalization of menin with neurofilament, which indicated neurite specific localization of menin (see Figure 3Ai–vi,B, *p* < 0.0001, *n* = 19 one-sample t-test, t = 41.39, *p* < 0.0001, *n* = 12; see Appendix A). (see Figure 3B) Cytofluorogram plotted between α-C-terminal menin and neurofilament indicated a high degree of colocalization and thus confirmed the neuron specific localization of α-C-terminal menin protein.

### 3.4. Astrocytes—Specific Expression of α-C-terminal Menin in Mouse Brain (In Situ) and (In Vitro)

We next sought to determine if menin protein was also expressed in astroglial cells. To address this question, we performed IHC and ICC assays to label mouse brain slices and glial cultures with glial fibrillary acidic protein (GFAP) (astroglial cell marker) and α-C-terminal menin, α-N-terminal menin epitope antibody. Our results from confocal high-resolution images exhibited significant colocalization of α-C-terminal menin (Figure 4Ai–iii) and α-N-terminal menin (Figure 4Bi–iii) with GFAP demonstrating astrocytes specific localization of menin (see Figure 4A,B, *p* < 0.0001, *n* = 19 one-sample t-test, t = 41.39, *p* < 0.0001, *n* = 12; see Appendix A). Cytofluorogram plotted between α-C-terminal menin and GFAP indicated high degree of colocalization and therefore confirmed astrocyte specific localization of α-C-terminal menin (see Figure 4C). Taken together, these data demonstrate that menin is not only localized to neurons but also non-neuronal cells.

### 3.5. Menin Expression Was Highly Localized to Telencephalon and Diencephalon during Embryonic Developmental Period in Mouse Brain

We next used IHC to characterize fluorescence intensity in order to deduce menin protein expression in embryonic mice brains. It is important to note that during early developmental stages, it was although difficult to decipher the precise demarcation of various brain regions, but the areas were carefully selected on the basis of information contained in the brain atlas [18] which reveals their eventual designations. Because C-menin was exclusively and predominantly expressed in neurons, and its role in both developing and adult brain is yet to be deduced, we thus focused on this protein throughout our developmental analysis. Brain sections obtained from stages E12-5-E18 were fixed and processed for C-menin antibody staining (Figure 5Ai–vi).

Our IHC data revealed that menin expression was highest in putative diencephalon and telencephalon regions during the embryonic periods E12.5 (Appendix A) and E15 (See Figure 5B, Appendix A, *p* < 0.0001, *n* = 13 one-way ANOVA followed by post-hoc Tukey test; see Appendix A), moderately expressed in metencephalon, whereas its expression was the lowest in the mesencephalon. These data thus suggest that C menin is likely differentially and systematically expressed in various brain regions at different embryonic stages.

We next continued to monitor menin distribution patterned and observed some dramatic shifts in its fluorescent intensity at E18. A remarkable increase in menin fluorescence intensity was observed compared to earlier embryonic stages of the mouse brain (see Figure 5B, Appendix A, *p* < 0.0001, *n* = 13 one-way ANOVA followed by post Tukey test; see Appendix A). This distribution exhibited significantly strong signal in telencephalon (Appendix A; *n* = 13 *p* < 0.0001, one-way ANOVA followed by post Tukey test; see Appendix A), as well as diencephalon regions respectively (Appendix A; *n* = 13 *p* < 0.0001, one-way ANOVA followed by post-hoc Tukey test; see Appendix A). At this developmental stage, the telencephalon, hippocampus and cortex exhibited more robust localization of menin protein, with there being the highest expression in the dentate gyrus and CA1 regions of the hippocampus (Appendix A), neocortex, frontal cortex, cingulate cortex and moderate expression in parietal cortex and insular cortex (Appendix A).

Within the diencephalon and thalamus exhibited a strong and ubiquitous signal, specifically in the nuclei of dorsal and ventral thalamic regions (Appendix A). Whereas moderate menin expression was observed in the hypothalamus (Appendix A). Within the metencephalon, moderate menin fluorescence intensity was detected, (Figure 5B, Appendix A
*p* < 0.0001, *n* = 13 one-way ANOVA followed by post-hoc Tukey test; see Appendix A) whereas mesencephalon exhibited very low to no immunoreactivity. Taken together, menin protein at E18 (See Appendix A)was predominantly localized to the telencephalon and diencephalon, whereas its expression was moderate in the metencephalon and low to no expression was observed in mesencephalon. Taken together, these data demonstrate that indeed menin’s fluorescence intensity gradually increased overtime throughout various embryonic brain development stages, suggesting a shift in its expression and distribution patterns with there being the strongest signals in diencephalon and telencephalon, moderate in metencephalon and lowest in mesencephalon.

### 3.6. Menin Exhibited Highest Expression in Thalamus, Hippocampus, Cortex and Cerebellum in Both P10 and Adult Mouse Brains

After establishing the spatio-temporal patterns of menin at various embryonic stages in a mouse brain, we next asked what happens to its distribution order after birth; does it increase, decrease or is diminished altogether? To answer the above questions, we labeled P10 mouse brain slices (16µm thick) with menin antibody as described above and imaged it at 40X magnification using the VS120 brain scanner. It is important to note that in order to mitigate any bias related to changes in tissue size and background fluorescence, we quantified fluorescence in specific regions of interest (ROI) of similar size and normalized the readout with respect to total brain area. The microscopic settings including the exposure time of filters were kept constant throughout the experiment as well.

Intriguingly, we detected no significant difference in the overall fluorescence intensity of menin protein from E18 to P10 (see Appendix A, *p* > 0.05, *n* = 13 one-way ANOVA followed by post Tukey test; see Appendix A). We did however notice a dramatic increase in menin immuno-fluorescence from P10 to an adult mouse brain (see Appendix A, *p* < 0.01, *n* = 13 one-way ANOVA followed by post-hoc Tukey test) and from E18 to an adult brain (see Appendix A, *p* < 0.0001, *n* = 13 one-way ANOVA followed by post Tukey test). Although menin protein localization increased in P10 mouse brains as compared with early embryonic ages (see Appendix A, *p* < 0.0001, *n* = 13 one-way ANOVA followed by post-hoc Tukey test), the protein distribution, however remained fairly similar to that of just before birth. Significantly higher localization was observed in telencephalon and diencephalon, whereas its intensity was moderate in the metencephalon and almost non-existent in mesencephalon.

Having established menin’s pattern of localization in postnatal mouse brains (see Appendix A), we next asked how these results would fair when compared with a fully-grown adult mouse brain. In this context, we found substantial increases in the localization of menin protein in distinct regions of the adult mouse brain, which mainly included cortex, thalamus, hippocampus and cerebellum and hypothalamus (Figure 6Ai–iv,B; see Appendix A).

Within the telencephalon, hippocampus and cortex, we observed strong menin fluorescence intensity (see Figure 6B, *p* < 0.0001, *n* = 13 one-way ANOVA followed by post-hoc Tukey test; see Appendix A), with robust localization in dentate gyrus, CA1, CA2 and CA3 regions of hippocampus (panel S5Aa,d), neocortex, frontal cortex, cingulate cortex, subventricular zone (panel S5Ab,e), and low localization in parietal cortex and insular cortex (panel S5Ag). Within the diencephalon, the thalamus exhibited strongest expression of menin (see Appendix A, *p* < 0.0001, *n* = 13 one-way ANOVA followed by post-hoc Tukey test; Appendix A) with high localization in nuclei of dorsal, central and ventral thalamic regions (panel S5Ac), whereas moderate expression was observed in the hypothalamus region (panel S5Ah). Within the metencephalon, the cerebellum exhibited moderate menin fluorescence intensity in the adult mouse brain (panel S5Af).

Together, the above data demonstrate that menin protein’s fluorescence intensity was the lowest at the early embryonic stage, which gradually increased through various developmental stages to later in the adult mouse brain. The spatio-temporal distribution of menin thus exhibited distinct localization in specific regions of telencephalon and diencephalon and metencephalon, whereas its expression was low to almost non-existent in the mesencephalon.

Taken together, the above data demonstrate that indeed menin is expressed throughout the developing mouse brain, albeit localized differentially and selectively to distinct regions. We also demonstrated for the first time that the cleaved C menin fragment is primarily localized not only to extrasomal neuronal regions but also the astrocytes. To decipher its potential developmental roles, we next asked the question if menin were to be potentially involved in regulating synaptic structures and function, then its expression patterns would likely coincide with other synaptic proteins.

### 3.7. Menin Was Colocalized with Both Pre- and Postsynaptic Protein in a Whole Mouse Brain

After demonstrating the presence of menin in the mouse brain through various developmental stages, we next sought to determine if it co-localized with pre- and/or postsynaptic proteins or both. Mouse primary hippocampal cultures and brain slices were prepared; of particular interest to us were pre- and postsynaptic markers such as Synaptotagmin (SYT-1) [19] and Postsynaptic density 95 protein (PSD-95) respectively. To confirm the colocalization regions of menin with each pre- and/or postsynaptic protein, we used IMAGE J, JACOP plugin to quantify the degree of colocalization (DOC) between each pair of relevant proteins. As described earlier, all values above 0.5 indicated strong colocalization (see Figure 7Ai–iii,Bi–iii).

We first sought to determine whether menin protein colocalized with SYT-1 and/or PSD-95 in the E12.5 mouse brain. Our data show that DOC between menin and SYT-1 was (0.42) significantly below the deduced value of 0.5 (Figure 7C; one-sample t-test, t = 4.91, *p* < 0.001, *n* = 12), which shows no colocalization between the proteins. Similarly, the DOC between PSD-95 and menin was (0.36), which was also below 0.5 (Figure 7C; one-sample t-test, t = 6.731, *p* < 0.0001, *n* = 12). These data thus conclude that the menin protein does not colocalize with either SYT-1 or PSD-95 in the E12.5 mouse brain.

Interestingly, in E15.5 brain slices, we observed significant colocalization of menin with SYT-1 (0.73) (Figure 7C; one-sample t-test, t = 33.66, *p* < 0.0001, *n* = 12). A strong correlation was also observed between menin and PSD-95 (0.812) (Figure 7C; one-sample t-test, t = 30.27, *p* < 0.0001, *n* = 12). This trend of high colocalization continued throughout later embryonic stages, where menin protein exhibited high DOC with SYT-1 and PSD-95 in E18 mice brain slices (See Appendix A).

We next asked the question, whether menin continues to colocalize with SYT-1 and PSD-95 after birth. To answer this question, we labeled P10 mouse brain slices with menin and SYT-1 and/or PSD-95 and found a gradual decrease in colocalization of menin with both pre- and postsynaptic protein; SYT-1 and PSD-95 respectively in P10 mouse brain slices compared to E18 brain slices (Figure 7C; one-sample t-test, t = 33.66, *p* < 0.0001, *n* = 12, see Appendix A).

Having established the colocalization patterns of menin with both SYT-1 and PSD-95, which were attenuated at the P10 mouse brain, we next investigated the colocalization patterns of menin with SYT-1 and/or PSD-95 in the adult brain. Specifically, we asked the question: does the DOC between these proteins increase, decrease or remain the same in a fully-grown adult mouse brain? To address this question, we prepared adult mice brain slices and stained them for all three proteins of interest. Our data revealed that menin’s colocalization with SYT-1 was significantly high (See Figure 7Biii) (0.896) (Figure 7C; one-sample t-test, t = 41.39, *p* < 0.0001, *n* = 12) in the adult brain, which was even higher than menin’s DOC with PSD-95 (See Figure 7Aiii) (0.820) (Figure 7C; one-sample t-test, t = 31.31, *p* < 0.0001, *n* = 12). The multiple comparison between DOC of menin with SYT-1 and PSD-95 in each age group was determined using Sidak’s multiple comparison test (See Appendix A, which concluded that there indeed existed a significant difference between the colocalization of menin with SYT-1 and PSD-95 at ages E12.5, E15.5 and the adult mouse brain. However, the DOC between menin and SYT-1 and PSD-95 was found to be insignificant at E18 and postnatal day 10 periods. In summary, we provided here a direct evidence in a whole mouse brain that menin indeed colocalizes with both pre- and postsynaptic protein (SYT-1 and PSD-95) and that this colocalization is patterned over embryonic, to postnatal period to an adult mouse brain.

Brain slices are a good in situ model as they allowed us to study the brain in its more original form, however, due to the anatomical complexity of multiple cell layers, coupled with glia labeling, they limited our ability to be more specific and precise at the level of individual axons and dendrites. We thus resorted to mouse hippocampal neuronal cultures using high resolution confocal microscopy. We fixed these cultures on different days in vitro (DIV) such as DIV4, DIV6, DIV7, DIV10, DIV 12 (which represents different stages of development compared to in situ, see Appendix A) and labeled the cultures with antibodies identifying SYT-1, PSD-95 and menin C- terminal (See Figure 8Ai–iii,Bi–iii).

Interestingly, our in vitro data also exhibited significantly high DOC of menin with SYT-1 on DIV4, which then gradually increased from DIV 6 to DIV12 (see Figure 8C; see Appendix A). This trend of gradual colocalization of SYT-1 and menin, was not observed with PSD-95 and menin as the DOC between PSD-95 and menin remained fairly high throughout all DIVs (see Appendix A). The most significant colocalization difference of menin with SYT-1 and PSD-95 was however, observed on DIV6, where menin DOC with PSD-95 was (0.839) significantly higher than its colocalization with SYT-1 (0.56) (Figure 8C, Sidak’s multiple comparison test, *p* < 0.0001, *n* = 18). The colocalization between menin and SYT-1 (See Figure 8Bi–iii), and menin and PSD-95 (See Figure 8Ai–iii) was seen in the boutons along the neurites, and also in the cytoplasm of the hippocampal neurons. Our in situ and in vitro data thus strongly suggest that menin shows a high degree of colocalization with both SYT-1 and PSD-95 from later embryonic stage (both in vitro and in vivo) to a fully mature adult brain. Taken together, these data demonstrate that menin indeed co-localizes with both pre-and postsynaptic proteins—with there being a stronger association with PSD-95, suggesting that these two proteins may function post-synaptically.

### 3.8. Menin Expression Was Significantly Aligned Along Excitatory versus Inhibitory Receptors in an Adult Mouse Brain Topography

The above data revealed a stronger correlation between menin, PSD-95 and SYT-1 in an adult mouse brain. To determine the phenotype of neurons where menin may function as a synapse regulator, we next sought to determine whether its expression patterns were confined to excitatory glutamatergic or the inhibitory, GABAergic neurons. To determine this, we employed adult mouse brain slices and stained them with metabotropic glutamatergic receptor 1 (mGluR1) [20], inotropic glutamatergic receptor AMPA subunit GluR1 [21] with receptor specific antibodies. Specifically, we labeled excitatory receptors and metabotropic GABAergic GABA B [22] and ionotropic GABAergic GABA A [21] with antibodies generated against inhibitory GABAergic receptors (see Figure 9Ai–iii,Bi–iii).

Our spatio (Figure 9C) and temporal data (Figure 9D) showed a high degree of menin colocalization with GABA-B receptors in the early embryonic periods (see Appendix A). This colocalization was, however, significantly reduced after birth to a fully-grown mouse brain (see Appendix A) (Figure 9D, one-sample t-test; *p* < 0.0001, *n* = 13). This pattern was however reversed in the case of mGluR1, where it showed low colocalization during early embryonic days, which gradually increased during late development (see Appendix A) (Figure 9D, one-sample t-test; *p* < 0.0001, *n* = 13). Similar trends of low to high colocalization were observed between GluR1 and menin, whereas high to low colocalization between menin and GABA A was noted (see Appendix A) (Appendix A one-sample t-test; *p* < 0.0001, *n* = 13).

As shown above, menin was found to be predominantly localized to the thalamus, hippocampus, cortex and cerebellum; we next investigated the DOC between menin, GABA-B and mGluR1 in these specific regions of the adult mouse brain. The brain slices labeled with their respective antibodies were imaged using confocal microscopy at 60X magnification to acquire high resolution images.

Our results demonstrate that menin indeed exhibited high DOC for mGluR1 in hippocampus, cortex (see Figure 9B), thalamus and cerebellum (see Appendix A) (Figure 9C, one-sample t-test; *p* < 0.0001, *n* = 14). Whereas menin DOC with GABA-B receptors was significantly less (see Appendix A–e) (see Figure 9A) compared to that of mGluR1 in all above-mentioned regions (Figure 9C, two-way ANOVA followed by Sidak’s multiple comparison test, *p* < 0.0001, *n* = 14, see Appendix A), suggesting that menin may primarily be functional at the excitatory synapses.

### 3.9. Menin Colocalizes with Nicotinic, Cholinergic Receptors (nAChR) Specific Subunits α5 and α7 in a Spatiotemporal Manner

Previous studies have shown that [6] menin regulates nAChR specific synaptic function but a direct correlation between these proteins in a developing brain has not yet been demonstrated.

Amongst the known functional subtypes of nAChRs [23], the most prevalent subtypes are α-bungarotoxin (α-BTX) sensitive α7 subunit [24] and its perturbations have been corroborated with neurodegenerative diseases such as Alzheimer’s [25,26]. Menin has previously been shown by us to regulate the transcription of α5 specific nAChRs [6]; we therefore sought to determine if menin protein colocalizes with these receptors in a spatiotemporal manner. To begin with, we used IHC assay to label mouse brain slices with α5 nAChR antibody and fluorophore-tagged α-BTX to label α5 and α7 nAChRs respectively (see Figure 10Ai–iii,Bi–iii). The brain slices labeled with various antibodies were imaged using confocal microscopy at 60X to acquire high resolution images.

We first sought to determine whether menin colocalizes with α5 nAChRs in whole mouse brain slices (in situ). Our data from the early embryonic stage such as E15.5, exhibited significant colocalization between menin and α5 nAChRs (See Figure 10C, one-way ANOVA followed by post-hoc Tukey’s test, *p* < 0.001, *n* = 15, see Appendix A). We next asked if this menin colocalization with α5 nAChRs continued post birth. P10 brain slices were prepared which revealed significantly high co-localization between the two (see Figure 10C, one-way ANOVA followed by post-hoc Tukey’s test, *p* < 0.001, *n* = 15, see Appendix A). Having established that menin colocalizes with α5 nAChRs in postnatal period, we next wanted to examine if menin and α5 nAChRs continue to be colocalized in an adult mouse brain or if this colocalization was limited to the postnatal period. Our adult mouse brain slice results revealed dynamic and strong DOC of menin with α5 nAChRs (see Figure 10C, one-way ANOVA followed by post-hoc Tukey’s test, *p* < 0.001, *n* = 15, see Appendix A). The colocalization of menin with α5 nAChRs in an adult mouse brain was also specific to four distinct regions where menin’s expression was reported to be the highest (as mentioned earlier), DOC in adult brain hippocampus (0.888), thalamus (0.899), cortex (0.857) and cerebellum (0.873) (see Appendix A).

We next sought to determine whether menin also colocalizes with α7 nAChRs in an embryonic mouse brain. Interestingly, our data from the early embryonic stage, such as E15.5 exhibited significant menin colocalization with α7 nAChRs (See Figure 10D, one-way ANOVA followed by post-hoc Tukey’s test, *p* < 0.001, *n* = 15, see Appendix A). We next asked whether this colocalization between menin and α7 nAChRs continues after birth or if it stabilizes at some point. We thus prepared P10 brain slices and saw significantly higher co-localization between the two (see Figure 10D, one-way ANOVA followed by post-hoc Tukey’s test, *p* < 0.001, *n* = 15, see Appendix A). Having established that menin colocalizes with α7 nAChRs in the postnatal period, we next investigated whether menin and α7 nAChRs continue to colocalize in an adult mouse brain as well, or if this colocalization was limited only to the postnatal period. The data from adult mouse brain slices showed dynamic and strong DOC between menin and α7 nAChRs (See Figure 10D, one-way ANOVA followed by post-hoc Tukey’s test, *p* < 0.001, *n* = 15, see Appendix A). The colocalization of menin with α7 nAChRs in an adult mouse brain was also specific to four distinct brain regions whereas its expression was found to be the highest as mentioned earlier. This augmented DOC in the adult brain was observed in hippocampus (0.914), thalamus (0.868), cortex (0.861) and cerebellum (0.856) (Appendix A).

In summary, menin exhibited colocalization with α5 and α7 nAChRs (see Figure 10Ai–iii,Bi–iii) in a mouse brain from early on during development to an adult brain. The augmented DOC was also specific to four distinct regions in a whole mouse brain.

## 4. Discussion

Recent studies have shown that genes that are thought to play specific developmental or regulatory roles in adult animals may in fact serve a myriad of other functions not previously ascribed to them [27]. An example of this would be *MEN1* gene and its encoded protein menin, which has previously been shown to function as a tumor suppressor [8]. Even though menin’s function in the cancer field [28] has been extensively studied, its involvement in the CNS from synapse formation [5,6,11] to learning and memory [3] in the adult animals was a surprising find. Notwithstanding its involvement in myriad brain functions, however, neither the morphological evidence for menin’s spatio-temporal patterns throughout various developmental stages nor its co-localization with other elements of synaptic machinery have yet been described. This lack of evidence may owe its existence to the fact that MEN1 knockout is lethal so its precise roles in brain development could not be reliably examined through a loss of function approach. Moreover, the complexity of the intact brain coupled with menin expression in astroglia makes it even more difficult to tease apart its functions at various individual developmental stages. Regarding menin expression in non-neuronal cells, a recent study has demonstrated that menin deficiency leads to depressive-like behavior in mice, and that these deficits involve astrocyte-mediated neuronal inflammation [29]. Whereas the precise role of glial menin remains elusive, our characterization of its distributions patterns in neurons coupled with coeval morphological juxtaposition with other elements of the synaptic proteins, shed significant light on its potential role in synaptic organization and function throughout development to adulthood.

We have also demonstrated here for the first time, that C and N menin fragments are expressed in both developing and adult brains, lending credence to the notion that MEN1 gene is indeed functional in various brain regions during early development and also in the adult brain. What role/s might these fragments be playing in a developing brain when various different brain regions are not yet clearly defined? An answer to this question may reside in the fact that menin, the product of MEN1, is shown to be a multifunctional, scaffold protein that regulates gene expression and cell signaling in either a negative or positive manner depending on the activity state of various second-messenger systems [8]. Ultimately, these interactions with a multitude of proteins having diverse functions may serve to transduce extracellular signals into meaningful and coordinated intracellular events. Functional repurposing of tumor suppressor gene function is commonly seen in synaptic development and regulation, perhaps the best understood example of this is the (DLG) family of tumor suppressors (e.g., post-synaptic density-95), which mediate glutamatergic synaptogenesis via the postsynaptic clustering of AMPA receptors involving PDZ domain protein-protein interactions [30]. Menin, however, exhibits no identified structural motifs or significant regions of homology to any known proteins, and as a result, we can deduce very little about the mechanisms underlying its developmental functions. Recent studies have, however, shown that menin through its interactions with p35 and cdk5 kinase cascades regulates dendritic growth which was perturbed after hippocampus specific MEN1, conditional knockout [3] leading to learning and memory deficit. It therefore stands to reason that menin may interact with p35-cdk-5 or other signaling molecules to regulate various functions during neuronal differentiation, proliferation, migration, neurite outgrowth and neuronal network assembly. However, its precise roles in these developmental steps await further analysis and characterization.

During neurogenesis, neurons are generated from neural stem cells [31], and this process occurs from early embryonic developmental stages to postnatal period [32,33] in a mouse brain. Our data shown here are thus consistent with the postulate that menin’s localization in the ventricular-subventricular zone (SVZ) as well as the sub-granular zone (SGZ) of dentate gyrus (DG) is indicative of its involvement throughout the embryonic period. As SVZ and SGZ of DG are main neurogenic niches in the mammalian brain [34], we thus speculate that menin may likely be playing a role in neurogenesis in the embryonic period of the mouse brain [35]. This postulate is consistent with other studies where menin was found to be a key player during the development and maintenance of multiple normal tissues [4]. It has also been reported in the literature that Notch [36], Wnt [37] and Shh [38] cascades are important regulatory signaling pathways known to play important roles during neurogenesis [33]. As menin has been shown to play significant roles in regulating signaling pathways such as BMP [39], Wnt [40] and hedgehog [41], these studies support our postulate further that it may also be involved in neurogenesis—at par with its expression patterns in distinct regions throughout the embryonic period as demonstrated earlier. Whereas, we have demonstrated here that menin expression is specific to SVZ and SGZ regions during embryonic period. What potential roles it may play there, however, still needs to be determined experimentally. The data presented in this study does, nevertheless, underscore the importance of genes, such as *MEN1*, which were thought initially to be the key players in regulating tumorigenesis, now they have turned up to play other myriad functions.

In the context of CNS development, following neuronal differentiation [42] and axonal migration [43], neurons begin to form synapses with their potential partners. This process of synaptogenesis begins later in embryonic development and continues throughout life [44]. The formation and consolidation of synapses require the expression of specific pre- and postsynaptic proteins at precise time frames and at appropriate locations in order to ensure the establishment of neuronal circuits that are behaviorally relevant [30,45,46]. Our study thus demonstrates that menin protein is indeed expressed in early embryonic stages where it exhibits high colocalization with the presynaptic protein SYT-1 as well as postsynaptic protein PSD-95, suggesting that its roles ought to be coordinated synergistically with both pre- and postsynaptic elements of the synaptic machinery [47]. Menin’s high degree of co-localization with PSD-95 tempts us to designate this MEN1 encoded protein as a “postsynaptic maker”. The interdependence of these two “postsynaptic” proteins would however still need to be demonstrated, further involving their individual experimental perturbations at various developmental stages. Parenthetically, a high degree of co-localization of both menin and PSD-95 with SYP may also elude towards the possibility that either menin, PSD-95 or both may serve as an inductive signal to translocate this presynaptic protein in juxtaposition to postsynaptic elements. This would also need to be demonstrated experimentally, but our findings do, nevertheless, indicate that menin could potentially be playing a significant role in either the formation of synapses or at the least targeting postsynaptic proteins to specific synaptic sites—either directly or via its interactions with PSD-95. Together these two postsynaptic proteins may thus serve as an inductive signal to recruit SYT-1 to specific presynaptic sites through yet to be defined cell–cell signaling mechanism. Previous studies support our reasoning where menin’s role in synapse formation [11], function [6] and synaptic plasticity [9] has been elucidated. Specifically, Getz et al. showed C-menin to be localized in perinuclear, cytoplasmic and in neuronal processes. Consistent with these findings, our current in situ study demonstrated specific localization of C and N fragments in the distinct regions of the mouse brain. Since previous studies have shown menin to be cleaved by Ca^2+^ activated Calpain enzyme requiring both trophic factors and electrical activity [11], and because differentiating neurons exhibit spontaneous Ca^2+^ oscillations [5], it is therefore reasonable to suggest that C-menin presence in neurite-less somata may also function to regulate transmitter receptor expression on the cell body. It is important to recognize that menin’s association with synaptic proteins such as SYT-1 and PSD-95 is essential not only for the development of synapses but also their maintenance; the perturbation of these proteins may likely underlie neurodegenerative diseases such as Alzheimer’s disease [48,49,50]. The precise interdependence and their interplay and the key regulatory roles of these essential synaptic proteins remains however to be determined but will likely shed significant light not only on the mechanisms underlying synapse formation but also synapse degeneration that is often seen in various disease conditions.

As aforementioned, menin is ubiquitously expressed during various embryonic stages, however we found its expression to increase during postnatal development in the mouse brain distinctly in the cortex, cerebellum, thalamus and hippocampus. This specificity of menin’s expression in these four regions is indicative of its tissue specific roles in each of these brain regions [8]. In the last decade, menin’s hippocampus specific role has been extensively studied [3,6] where it has also been implicated in learning and memory [3,51]. Notwithstanding, this elegant demonstration of menin’s roles, morphological evidence for its expression patterns in the brain, and more specifically its relationship with other synaptic proteins, has not yet been shown at any given developmental stage. Our study fills this gap by mapping menin protein in the hippocampus from development to a mature mouse brain, which provides morphological evidence for its involvement in learning and memory. Previous studies have shown that *MEN1* gene [2] is expressed in all brain regions throughout development—albeit becoming restricted to hippocampus only. Our data for menin are consistent with those finding, and we further suggest that the reasons for this continued menin expression may reside in the fact that whereas neurogenesis stops in all other brain regions, the hippocampus continues to generate new neurons even in adulthood. Thus, it is plausible that these newborn hippocampal neurons, which are also preferentially recruited in the learning and memory circuits [52], may require continued presence and function of menin. It is also important to note that our data, for the first time, has demonstrated high menin expression in the cerebellum and thalamus, thus underscoring its potential roles in other brain functions as well.

We demonstrated significant colocalization of menin with inhibitory GABAergic receptors in early development compared to glutamatergic receptors, which would be indicative of its role at excitatory synapses since the former are known to gate excitatory channels during early development [53]. Later in the adult mouse brain, the ratio of menin expression with excitatory receptors is significantly higher than inhibitory neurons, which suggests that it may likely be predominantly involved in excitatory versus inhibitory synaptic transmission. However, previous studies have shown that MEN1 knockdown did not affect the function of glutamatergic synapses [6] which implies that its affects may likely be indirect through modulatory synapses involving nAChRs [54], where it may enhance the efficacy of synaptic transmission between neurons involved in direct processing of learning and memory. This postulate, however, needs to be tested experimentally.

The data presented here provide evidence for colocalization of menin with nAChRs specific subunits α7 and α5 in an in situ mouse brain model. According to previous studies, during early development, nAChRs are expressed in the mouse brain [55] but most important, among the nAChRs are α7 subtypes [56], as they have vital roles in the development and function of various circuits in the brain such as the hippocampal circuity [57]. For instance, α7 specific nAChR signaling is known to promote the survival and maturation of neurons [58], along-with promoting synapse formation in the hippocampus [59]. As menin exhibited significant colocalization with α7 and α5 nAChRs extensively. Our data suggest that menin might be a crucial protein for the localization and function of these α7 and α5 receptors to specific synaptic sites. This notion would be in agreement with previous findings on the role of menin in coordinating the transcriptional upregulation of α5 and synaptic clustering of α7-nAChRs in mouse hippocampal neurons in vivo [6]. This study is, however, the first to demonstrate concomitant co-localization of three postsynaptic proteins—not only at developing, but also fully matured synapses: namely menin, PSD-95 and nAChRs. Previous research has highlighted cognitive deficits in Alzheimer’s disease (AD) associated with cholinergic dysfunction [60]. Additionally, perturbations of cholinergic signaling, generate a range of disorders associated with a diversity of neurological disorders and diseases including AD, schizophrenia and epilepsy [26,61] that are met in the clinic. Understanding the underlying mechanisms of how menin interact with these nAChRs receptors and mediate their actions will bring us a step closer to understanding the precise roles of these receptors in normal brain development, physiology, as well as NDD associated with the perturbed expression and function of nAChRs.

## Figures and Tables

**Figure 1 cells-10-01215-f001:**
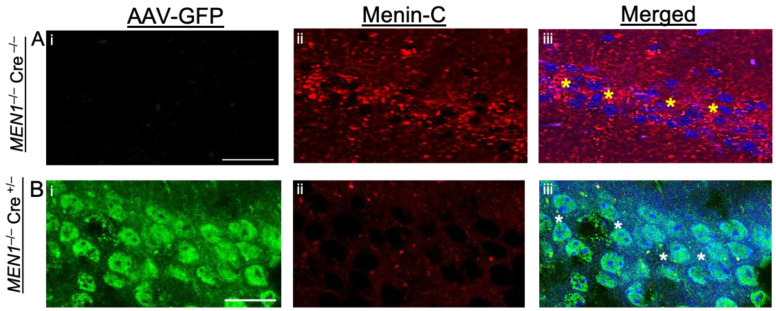
Neurons specific menin deletion in Cre activated, double floxed MEN1 homozygous mice confirmed the specificity of C-menin expression in the hippocampus. (**Ai**–**iii**,**Bi**–**iii**) IHC high magnification confocal image of an adult mouse hippocampal brain slice (*n* = 40 images, 2 independent samples, representative image). (**A**) Negative controls zoomed in shot of (CA1) labeled with α-C-terminal menin (**Aii**), (**Aiii**) show merged image. As shown, they are negative of any Cre expression (labeled with GFP; **Ai**), (*n* = 18 images, 2 independent samples, representative image). (**B**) Cre-AAV positive neurons (**Bi**) in CA1 region of mouse hippocampus depicts menin deletion in the perinuclear, cytoplasmic and synaptic regions as shown (B**ii**), (B**iii**) show merged image. Yellow asterisks, Cre negative neurons in CA1 region with robust menin expression, white asterisks, Cre positive neurons in CA1 region with low menin expression GFP labeled. Scale bar, (**A**) 25 μm, (**B**) 20 μm.

**Figure 2 cells-10-01215-f002:**
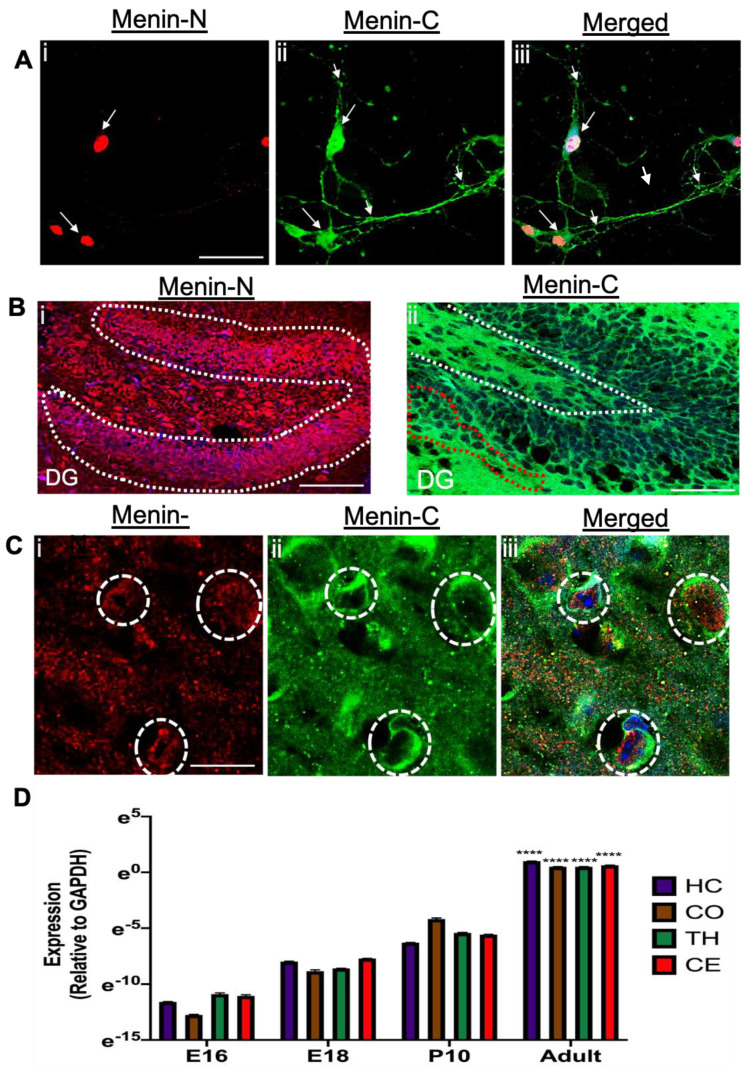
Hippocampus neurons labeled with α-C-terminal menin and α-N-terminal menin (in-vitro) and (in-situ) depicting that menin fragments are differentially localized in mouse neurons. (**A**) High magnification confocal image of cultured hippocampal neurons ROI at DIV 7(*n* = 18 images, 8 independent samples, representative image), adult mice hippocampal brain slices were labeled with α-N-terminal menin (**Ai**) α-C-terminal menin (**Aii**), (**Aiii**) shows merged channels. (**B**) As in (**A**), only depicting α-N-terminal menin (**Bi**) and α-C-terminal menin (**Bii**) localization in dentate gyrus (DG) hippocampal slice (*n* = 20 images, 12 independent samples, representative image). (**C**) High magnification confocal image of hippocampus slices depicting α-N-terminal menin (**Ci**) and α-C-menin termina (**Cii**), merged (**Ciii**) high expression in the neurons. Arrowheads, localization of N-menin in the nuclei with α-C-terminal menin in the perinuclear, cytoplasmic and synaptic regions as in (**A**) and (**B**) white dotted region represents N-menin expression in the nuclei, co-labeled with nuclear stain DAPI (**Ai**,**Bi**,**Ci**) red dotted regions depicts α-C-menin in the perinuclear and cytoplasmic regions respectively. (**C**) dotted circles represent individual neuron expressing both α-N-menin and α-C-menin. Scale bar, (**A**) 20 μm, (**B**) 25 μm (**C**) 5 μm. (**D**) Summary data, showing spatiotemporal fold changes in *MEN1* gene expression in mouse brain from E12.5 to an adult brain; specifically, in hippocampus, thalamus, cortex and cerebellum, relative to control GAPDH gene, determined by qPCR (*n* = 6, 3 independent experiments each, triplicate replicates). Asterisks, statistical significance between the adult MEN1 gene expression and other ages (one-way ANOVA followed by post-hoc Tukey test); **** *p* < 0.0001. See also Appendix A.

**Figure 3 cells-10-01215-f003:**
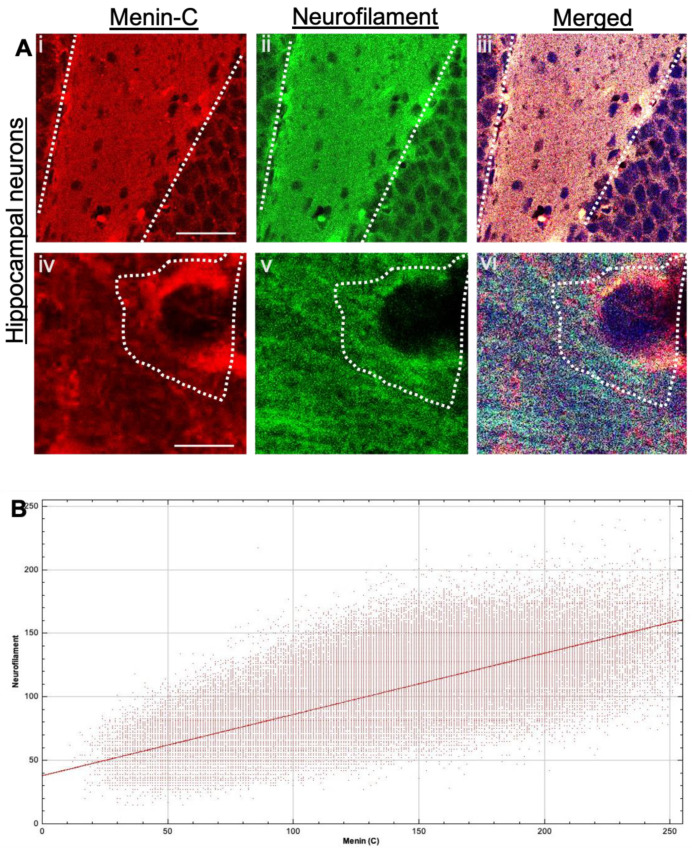
Neuron specific expression of α-C-terminal menin in mouse neurons (in-situ). (**Ai**–**vi**) IHC High magnification confocal image of a hippocampal brain slices (*n* = 20 images, 9 independent samples, representative image), mice hippocampal brain slices were labeled with α-C-terminal menin (**Ai**,**iv**) and neurofilament (neuronal marker) (**Aii**–**v**), (**Aiii**,**vi**) showing merged images respectively. (**Aiv**–**vi**) shows enlarged images of colocalization of α-C-terminal menin and neurofilament at a high magnification (single neuron). White dotted regions, co-localization of α-C-terminal menin in the synaptic regions with neurofilament. (**A**). Scale bar, (**Ai**–**iii**) 35 μm, (**Aiv**–**vi**) 7 μm. (**B**) Cytofluorogram plotted depicting high degree of colocalization between α-C-terminal menin and neurofilament using Pearson’s coefficient score. (ROIs: *n* ≥ 19 each from four independent samples; see Appendix A).

**Figure 4 cells-10-01215-f004:**
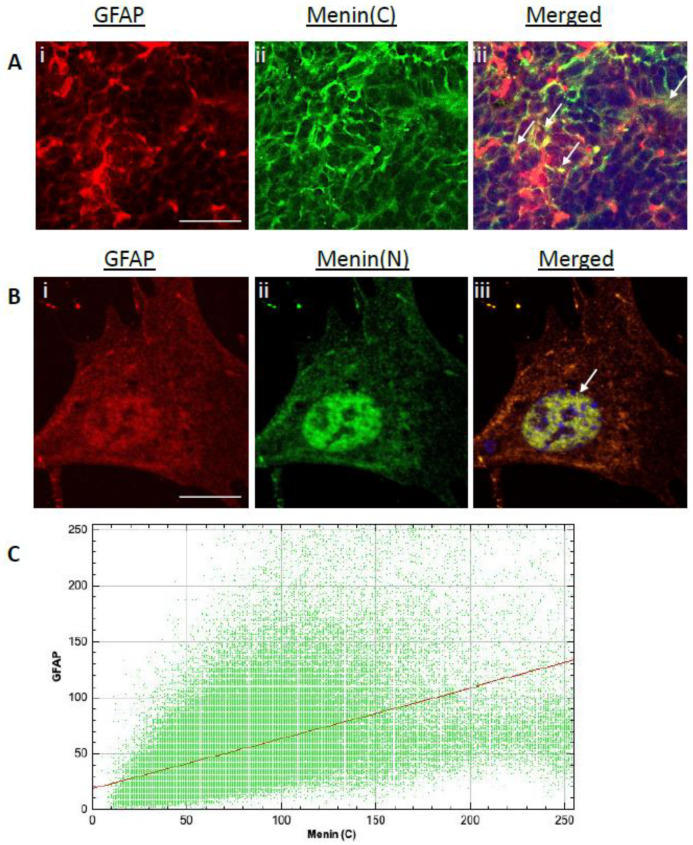
Astroglial specific expression of α-C-terminal menin in hippocampal brain slices (in situ) α-N-terminal menin in glial cultures (in vitro). (**Ai**–**iii**) IHC high magnification confocal image of a hippocampal brain slices (*n* = 20 images, 9 independent samples, representative image), mice hippocampal brain slices were labeled with α-C-terminal menin (**Aii**) and GFAP (astroglial marker) (**Ai**), (**Aiii**) showing merged images respectively. (**Bi**–**iii**) As in (**A**), only showing the two proteins in glial cultures using α-N-terminal menin (**Bii**) respectively. Arrowheads, co-localization of α-C-terminal menin with GFAP in astroglial cells. (**A**). Scale bar, (**Ai**–**iii**) 35 μm, (**Bi**–**iii**) 5 μm.( **C**) Cytofluorogram plot depicting high degree of colocalization between α-C-terminal menin and GFAP labeled astroglia using Pearson’s coefficient score. (ROIs: *n* ≥ 19 each from four independent samples; see Appendix A).

**Figure 5 cells-10-01215-f005:**
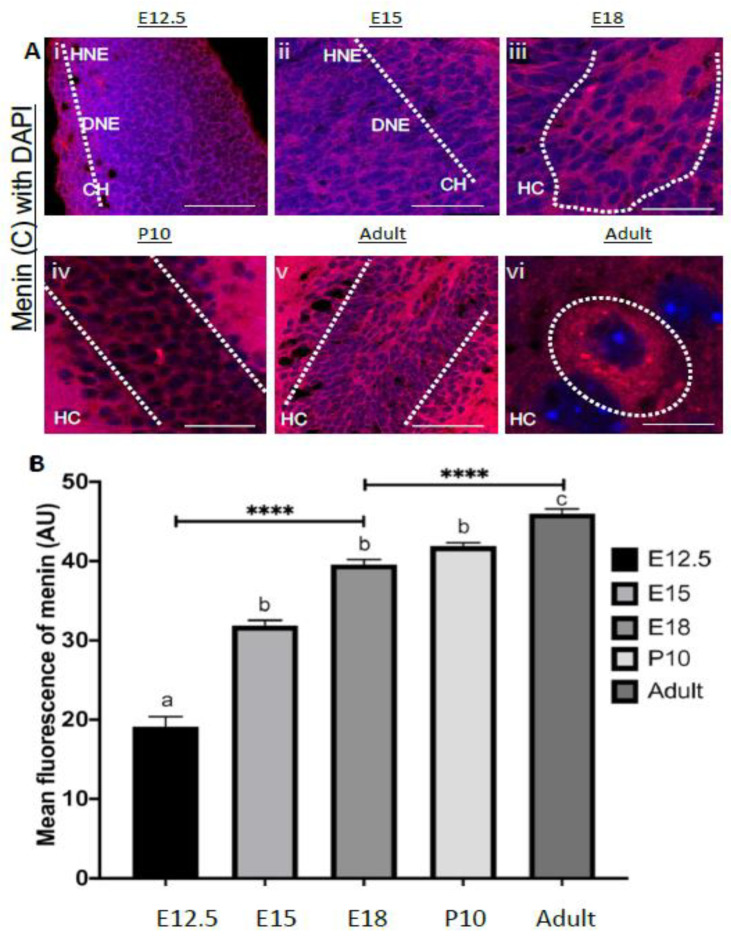
Spatiotemporal localization of α-C-terminal menin increases in mice hippocampal brain slices over period (**A**) High magnification confocal image of a hippocampal brain slices (*n* = 18 images, 9 independent samples, representative image), mice hippocampal brain slices from E12.5, E15, E18, P10 to adult ages were labeled with α-C-terminal menin (**Ai**–**vi**), vi shows α-C-terminal menin at a single neuron level. White dotted regions show localization of α-C-terminal menin in the perinuclear, cytoplasmic and synaptic regions, and increase in the α-C-terminal menin spatio-temporarily (**A**). Scale bar, (**Ai**) 25 μm, (**Aii**) 15 μm, (**Aiii**) 12 μm, (**Aiv**) 15 μm, (**Av**) 25 μm, (**Avi**) 2 μm. (**B**) Summary data, α-C- terminal menin from E12.5, E15.5, E18, P10 mouse brain to an adult mouse brain (ROIs: *n* ≥ 19 each from 8 independent samples; see Appendix A). Group (a) (significantly low expression) and group (b) (no significant difference within the group but significantly higher than and lower than c) group (c) (significantly highest expression). Asterisks, statistical significance (one-way ANOVA followed by post-hoc Tukey test); **** *p* < 0.0001.

**Figure 6 cells-10-01215-f006:**
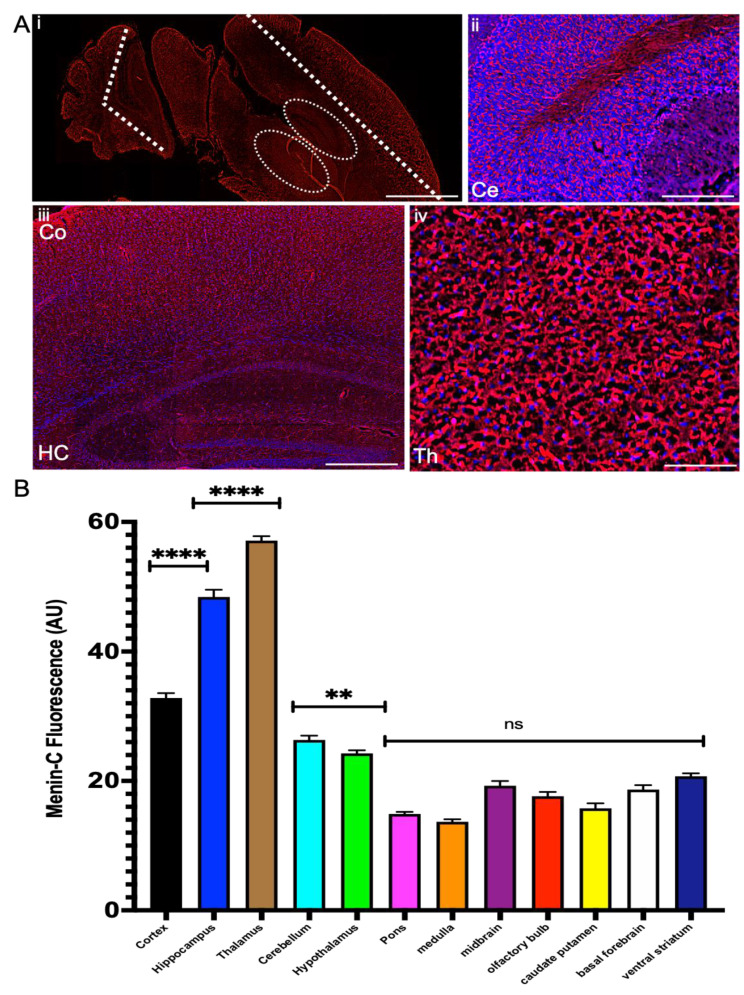
Menin protein expression in an adult mouse brain. (**A**). Menin protein distribution is represented on a sagittal section. (**Aii**–**iv**) zoomed in high magnification images. IHC characterization of menin protein in an adult mouse brain slice exhibits high localization in the metencephalon, cerebellum exhibited high menin fluorescence intensity in the adult mouse brain (panel **ii**). Within the telencephalon especially in the hippocampus (panel **Aiii**), in cortex (panel **Aiii**). Within the diencephalon, high localization in nuclei of dorsal, central and ventral thalamic regions (panel **iv**). Scale bar: (**Ai**). 300 μm, panel (**Aii**) 55 μm, panel (**Aiii**) 85 μm, panel (**Aiv**) 65 μm, panels. (**B**). Summary data showing normalized fluorescence intensity of α-C-terminal menin in different regions of adult mouse brain (ROIs: *n* = 13 each from 7 independent samples; see Appendix A). Asterisks, statistical significance (one-way ANOVA followed by post-hoc Tukey test); ** *p* < 0.01, **** *p* < 0.0001, ns ≥ 0.05.

**Figure 7 cells-10-01215-f007:**
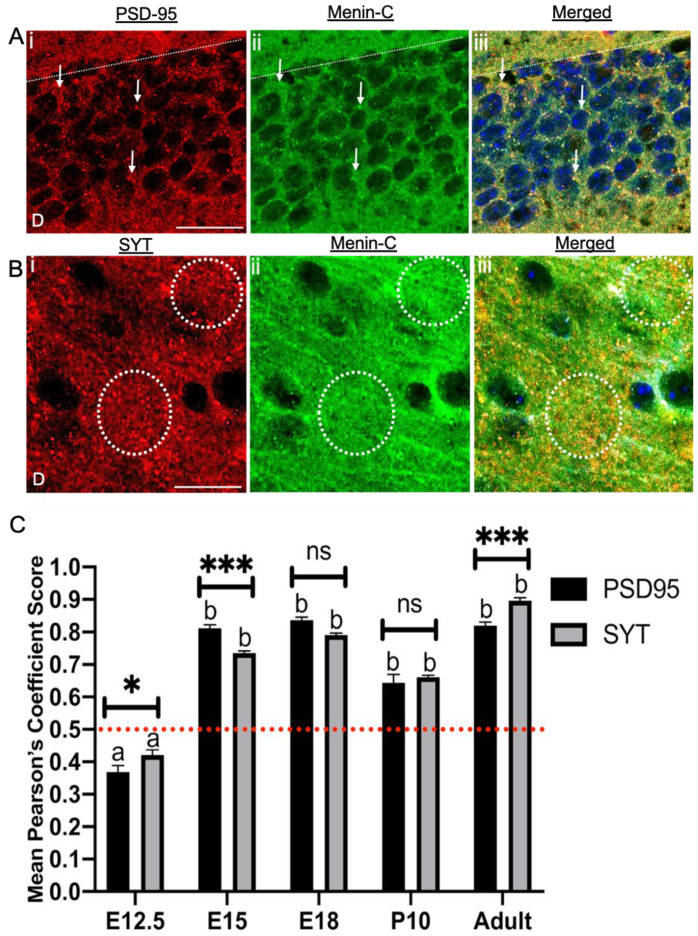
The C-terminal menin protein colocalizes with SYT-1 at presynaptic terminals and PSD-95 at post-synaptic terminals in whole mouse brain hippocampal slices. (**A**) High magnification confocal image of a synaptic ROI of an adult brain slice (*n* = 12 images, 3 independent samples, representative image), brains slices were labeled with PSD-95 (A**i**; postsynaptic marker), α-C-terminal menin (**Aii**), (**Aiii**) shows merged channels. (**B**) As in (**A**), only labeled with α-SYT-1 (B**i**; presynaptic marker) α-C-terminal menin (**Bii**), (**Biii**) shows merged channels. (*n* = 13 images, 7 independent samples, representative image). White dotted regions, colocalization of C-menin with PSD-95 and α-SYT-1 in (**A**) and (**B**) respectively; (**A**) and (**B**) at single neuron and puncta level. Scale bar, (**A**) 20 μm (**B**) 20 μm. (**C**) Summary data, Pearson’s co-efficient, temporal pattern showing degree of colocalization of Synaptotagmin (SYT-1; presynaptic), PSD-95 (postsynaptic) puncta with C-menin in E12.5, E15.5, E18, P10 and adult mouse brain. Red dotted line indicates the literature value of significant DOC; group a (no DOC significantly) and b (DOC significantly) (one-sample t-test). Asterisks, statistical significance (one-way ANOVA followed by post-hoc Tukey test); **p* < 0.01, *** *p* < 0.001, ns ≥ 0.05 See also Appendix A.

**Figure 8 cells-10-01215-f008:**
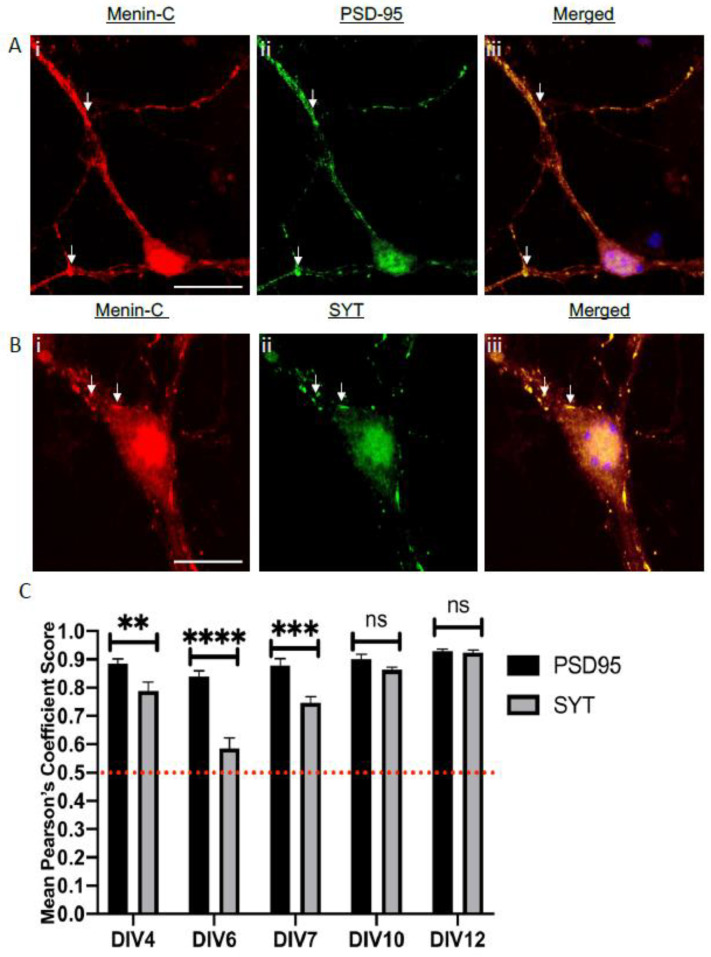
The C-terminal menin protein colocalizes with SYT-1 at presynaptic terminals and PSD-95 at post-synaptic terminals in hippocampal neuronal cultures. (**A**) High magnification confocal image of a synaptic ROI at DIV 7(*n* = 18 images, 5 independent samples, representative image), neuronal cell cultures were labeled with α-C-terminal menin (**Ai**), PSD-95 (**Aii**; postsynaptic marker), (**Aiii**) shows merged channels. (**B**) α-C-terminal menin, (**Bi**) labeled with α-SYT-1 (**Bii**; presynaptic marker) (**Biii**) shows merged channels (*n* = 22 images, 12 independent samples, representative image). Arrowheads, colocalization of C-menin with PSD-95 and α-SYT-1 in (**A**) and (**B**) respectively. Scale bar, (**A**,**B**) 10 μm. (**C**) Summary data, Pearson’s co-efficient, temporal pattern showing degree of colocalization of Synaptotagmin (SYT-1; presynaptic), PSD-95 (postsynaptic) puncta with C-menin in DIV 4, DIV 6, DIV 7, DIV 10 and DIV 12. Red dotted line indicates the literature value of significant DOC. Asterisks, statistical significance (one-way ANOVA followed by post-hoc Tukey test); ** *p* < 0.01, *** *p* < 0.001, **** *p* < 0.0001, ns ≥ 0.05 See also Appendix A.

**Figure 9 cells-10-01215-f009:**
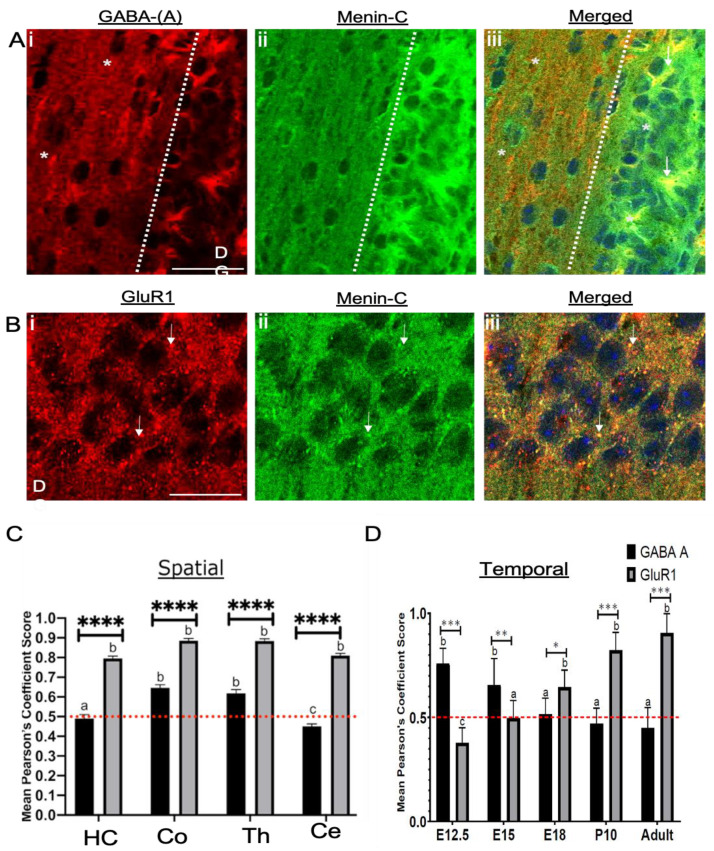
Menin protein expression patterns were significantly higher at excitatory versus inhibitory neurons in an adult mouse brain topography. (**A**) Confocal, high magnification image of a synaptic ROI of an adult brain slice (*n* = 14 images, 5 independent samples, representative image), brains slices were labeled with GABA A (**Ai**; inhibitory neuronal marker), α-C-terminal menin (**Aii**), (**Aiii**) shows merged channels. (**B**) labeled with GLuR1 (**Bi**; excitatory neuronal marker) α-C-terminal menin (**Bii**), (**Biii**) shows merged channels. (*n* = 14 images, 8 independent samples, representative image). Arrowheads, colocalization of C-menin with GABA A and GluR1 in (**A**) and (**B**) respectively; asterisks indicate no degree of colocalization. (**A**) and (**B**). Scale bar, (**A**) 35 μm (**B**) 18 μm. (**C**) Summary data, Pearson’s co-efficient, spatial pattern degree of colocalization of GABA A and GluR1 puncta with C-menin in distinct regions of an adult mouse brain. (**D**) Summary data, Pearson’s co-efficient, temporal patterns degree of colocalization of GABA A and GluR1 puncta with C-menin in E12.5, E15.5, E18, P10 and adult mouse brain mouse brain. Red dotted line indicates the literature value of significant DOC; group a (DOC significant precisely at 0.5), b (DOC significant) and group c (no significant DOC) (one-sample t-test). Asterisks, statistical significance (one-way ANOVA followed by Sidak’s comparison test); * *p* < 0.01, ** *p* < 0.01, *** *p* < 0.001, ***** p* < 0. 0001. See also Appendix A.

**Figure 10 cells-10-01215-f010:**
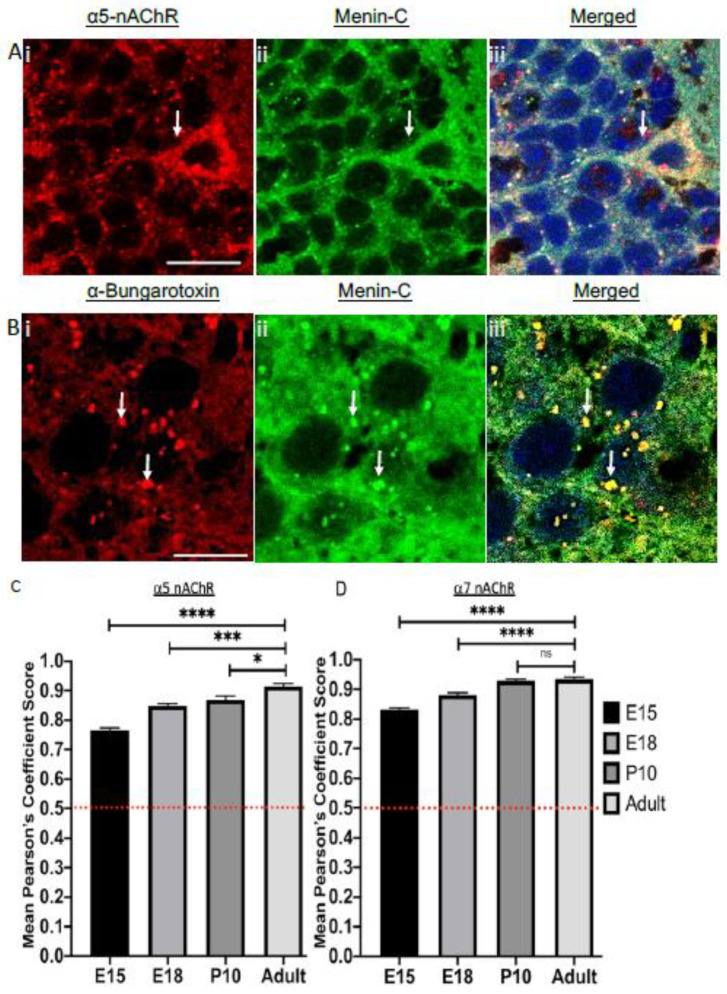
Menin protein colocalizes with nicotinic cholinergic receptors (nAChR) specific subunits α5 and α7 in a spatiotemporal pattern in a whole mouse brain. (**A**) Confocal high magnification image of a synaptic ROI of an adult brain slice (*n* = 15 images, 6 independent samples, representative image), brains slices were labeled with nAChR specific subunit α5 (**Ai**), α-C-terminal menin (**Aii**), (**Aiii**) shows merged channels. (**B**) labeled with α7 labeling α-bungarotoxin (**Bi)**, α-C-terminal menin α5 (**Bii**), (**Biii**) shows merged channels (*n* = 15 images, 6 independent samples, representative image). Arrowheads, colocalization of C-menin with α5 and α7 in (**A**) and (**B**) respectively. Scale bar, (**A**) 25 μm (**B**) 20 μm. (**C**,**D**) Summary data, Pearson’s co-efficient, temporal pattern degree of colocalization of α5 (nAChR specific subunit), α5 (nAChR specific subunit) respectively in puncta with C-menin in E12.5, E15.5, E18, P10 and adult mouse brain. Red dotted line indicates the literature value of significant DOC; (one-sample t-test). Asterisks, statistical significance (one-way ANOVA followed by post-hoc Tukey test); ns ≥ 0.05, * < 0.01, *** *p* < 0.001, ***** p* < 0.0001. See also Appendix A.

## Data Availability

The data presented in this study are available on request from the corresponding author.

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
