# Peer review of "Spatiotemporal Patterns of Menin Localization in Developing Murine Brain: Co-Expression with the Elements of Cholinergic Synaptic Machinery"

_cells, 2021, doi:10.3390/cells10051215_

Round 1

Reviewer 1 Report

The manuscript by Batloon et al., defines spatiotemporal localization of menin, a product of MEN1 gene,  in the brain, from the embryonic phase to adult.  The authors showed that menin expression goes from widespread in early development stages to highly localized in cognition centers in CNS. This study provides the first direct evidence for menin co-expression with nicotinic acetylcholine, glutamate, and GABA receptors, thus making point of the importance of menin in cholinergic neuronal network assembly. Overall, the studies are well-conducted and the data are of interest, especially data related to menin colocalization with pre-post synaptic markers as well as nAChR in the developing brain.

Comments: My overall suggestion to authors would be to pay more attention to figure legend and overall editing of the manuscript (see specific comments).

Comment 1: In this study, the authors discuss the presence of menin in non-neuronal cells, specifically in astrocytes. Was this just in the adult brain (Table S21B that author refers to couldn’t be found in the supplementary file)? Taking into account that microglia plays a key role in establishing neural circuits in the developing brain it would be interesting to test for the presence of menin in microglia (iba1) in the developing brain.

Comment 2: Authors should transfer some of the data from supplementary data to Fig 4. in the main text related to menin expression in different brain areas during embryonic development and adult brain. Spatiotemporal localization should be better present inside the main text.  

Comment 3: Authors in the discusion part made an overall good connection with menin colocalization data with α5 and α7 nAChRs and therefore their possible interaction and possible role in cholinergic disfunction. In that note to further solidify these, I would suggest to authors look into the possible interaction of α4β2 nAChRs subtype.

Specific comments:

P6, figure 1ii. – I would suggest to authors put an arrow to puncta formulations since they discuss their presence at line 231.

Line 264. – Should state – “in the hippocampus, cortex, thalamus and cerebellum (Fig 1D)” missing figure 1D, in this paragraph, it’s only pointing out to Table6S at Supplementary but not to figure in the main text.

Line 284 – Correct (D) Cytofluorogram to (B) Cytofluorogram

Line 292 – Table S21b… Couldn’t find that table in the supplementary file

Line 303 – Correct (D) to (B)…. This happens throughout every figure legend. I will not comment on it more. Please correct it.

P10, figure 4B.- Please clarify statistical significance related to letters above bars on this graph-it’s not clear.

Author Response

We are grateful to both reviewers for their constructive comments and the feedback that they have provided. Most issues raise were fair and this manuscript is now revised in light of both referee’s suggestions. The concerns raised by this referee are addressed bellow on a point-by-point basis.

Reviewer 1:

Comment 1: In this study, the authors discuss the presence of menin in non-neuronal cells, specifically in astrocytes. Was this just in the adult brain (Table S21B that author refers to couldn’t be found in the supplementary file)?

Response: The colocalization was deduced on embryonic day 16, embryonic 18, postnatal 10 and in adult brain. The summary data is now presented in table 19b in the supplementary document.

Comment: Taking into account that microglia play a key role in establishing neural circuits in the developing brain it would be interesting to test for the presence of menin in microglia (iba1) in the developing brain.

Response: We fully acknowledge the importance of microglia in the establishment of developing brain as suggested by the referee. This study was however, designed to focused primarily on menin and its specific role in neuronal circuits that underlie learning and memory in the hippocampus – particularly in neuronal circuits that invoke nicotinic synaptic machinery. Moreover, a recent study has already demonstrated that menin doesn’t colocalize with microglia and it’s function is specific only to astroglial cells therefore only astrocytes were included in our study.[1]

Comment 2: Authors should transfer some of the data from supplementary data to Fig 4. in the main text related to menin expression in different brain areas during embryonic development and adult brain. Spatiotemporal localization should be better present inside the main text.  

Response: As per referee’s suggestions, we have now added a new figure 5 in the main result section.

Comment 3: Authors in the discussion part made an overall good connection with menin colocalization data with α5 and α7 nAChRs and therefore their possible interaction and possible role in cholinergic disfunction. In that note to further solidify these, I would suggest to authors look into the possible interaction of α4β2 nAChRs subtype.

Response: The referee has raised an interesting point however our previous study [2] has shown that menin primarily plays an important regulatory function involving nAChRs subunit α5 and α7 - out of 9 possible other subunits that were investigated. Therefore, we primarily focused on these two receptor subunits. Moreover, the α4 β2 subunits have been shown to play role in neuropsychiatric disorders, including autism spectrum disorders, nicotine addiction[3] which was not the focus of our study.

Specific comments: 

P6, figure 1ii. – I would suggest to authors put an arrow to puncta formulations since they discuss their presence at line 231.

Changes made as per the suggestion.

Line 264. – Should state – “in the hippocampus, cortex, thalamus and cerebellum (Fig 1D)” missing figure 1D, in this paragraph, it’s only pointing out to Table6S at Supplementary but not to figure in the main text.

Changed accordingly.

Line 284 – Correct (D) Cytofluorogram to (B) Cytofluorogram

Changed

Line 292 – Table S21b… Couldn’t find that table in the supplementary file

We acknowledge this oversight, the table is now replaced with Table S19b

Line 303 – Correct (D) to (B)…. This happens throughout every figure legend. I will not comment on it more. Please correct it.

We acknowledge and the errors are now corrected throughout.

P10, figure 4B.- Please clarify statistical significance related to letters above bars on this graph-it’s not clear.

The statistical significance is now highlighted.

  1. Leng, L.a.Z.K.a.L.Z.a.H.C.a.G.Y.a.C.G.a.L.H.a.H.Y.a.W.D.a.S.M.a.X. Menin Deficiency Leads to Depressive-like Behaviors in Mice by Modulating Astrocyte-Mediated Neuroinflammation. Neuron (Cambridge, Mass.) 2018, 100, 551--563.e557.
  2. Getz A.M.a.F.X.a.F.V.a.R.P.a.N.I.S. Tumor suppressor menin is required for subunit-specific nAChR α5 transcription and nAChR-dependent presynaptic facilitation in cultured mouse hippocampal neurons. Scientific Reports 2017, 7, 1.
  3. Caton, M.a.O.E.L.M.a.B.F.J. The role of nicotinic cholinergic neurotransmission in delusional thinking. NPJ schizophrenia 2020, 6, 16--16.

Reviewer 2 Report

This work examines the distribution of menin in the murine brain along development. It is exclusively descriptive and any possible functions of the protein in the context of brain development and function are simply inferred from its brain region or cellular localization, or its co-localization with other proteins. While the results presented by the authors could have some interest to the scientific community, there are many issues that need to be addressed. 

One of these is the language used. I strongly advise the authors to have the text proofed by a native English speaker or at least someone experienced. There are errors that should not have been found in a submitted manuscript, such as saying in the abstract that their study “is the first to provide an incompetent (the synonym of inadept) analysis[...]”.

Introduction:

The introduction shows what is known in the field with relevance for the current study, but it is often written in an odd way. For example, stating the objectives as a long list of questions seems awkward.

Methods:

The methods are generally written with enough detail, but some parts are not clear, as listed:

  • The mouse strain should be spelled C57BL/6;
  • Neurons are dissociated, not detached;
  • “plated at low to medium density” is vague and actual density /mm2 or cm2 should be given
  • “Negative controls were performed to test the specificity of the antibodies”. Which controls specifically? NO detail are given and no results either, which seem to be part of a previous publication. Therefore, this data should either be shown or cited.
  • Using fluorescence intensity as AU units to quantify menin density without specifying the range of AU is not useful

Results:

Section 3.1 is not a results section and should just be omitted, since the antibodies were tested in a previous study.

One major problem I see with the data is that the immunohistochemistry data is not very convincing, since the antibodies seem to label almost everything and everywhere. A proper control using neurons from KO mice would be very welcome to demonstrate staining specificity, especially since both antibodies label more than 1 band in western blots, as shown previously by the authors. The authors do mention negative controls but these are not shown.

The N fragment of menin is found only in the nucleus, whereas the C fragment seems to be everywhere, including the nucleus (Fig 1B). However, the authors’ description is misleading since it makes no mention of menin C also being nuclear. Also, this data is similar to the one in their previous publication (Angela et al, 2017) and could probably be omitted altogether.

The authors also claim that the expression of menin is highest in the hippocampus, cortex, cerebellum and thalamus. However, no data is presented for any other brain regions. To make this claim, other brain regions should have been quantified and displayed, and a full brain section (coronal or sagittal) reconstructed from a mosaic of confocal images is lacking for the reader to appreciate the pattern of menin distribution in the brain. By only displaying magnified views of selected regions could show data bias.

Regarding the figure legends, confocal microscopy is referred to as “super high resolution confocal” microscopy. This does not exist and the scale bars show that “normal” confocal microscopy was used. Super resolution microscopy allows resolutions of 200 nm or under, which cannot be attained with confocal microscopy (not at least without special sample preparation, as in expansion microscopy)

Section 3.3 says that menin C fragment is specifically expressed in neurons, but section 3.4 says that it is specifically localized to astrocytes. It would appear that the protein is localized everywhere and this could be simplified by just saying that it is present in both neurons and astrocytes.

Data regarding the localization of menin expression during embryonic and postnatal development would also greatly benefit from a whole brain section, to show possible localized patterns in specific brain regions

The co-localization data with syt and PSD95 is extremely unconvincing. What do the authors mean by a “synaptic ROI”? Surely the authors don’t think that the area surrounded by a circle in fig 5 corresponds to synapses? The scale bar reads 20 um, which is the size of whole neurons, not synapses. The labeling appears to be intracellular, surrounding nuclei in what could be interneurons, but it’s unclear from what hippocampal region the images are fro. Also, the images in the red channel and green channels seem to have greatly different pixel sizes, so even if the labeling were of synapses, co-localization would be hard to interpret in such conditions. The authors should have used higher magnification images and performed colocalization analysis at the single puncta level. The authors should specify which syt isoform is being labeled, since some are postsynaptic and be aware that no syt is a de facto presynaptic marker. Bassoon or other presynaptic active zone protein would have been a better choice.

The data on excitatory vs inhibitory localization of menin in unconvincing. First, the authors fail to mention where in the brin they are imaging (it’s only referred to as synaptic ROI of an adult brain slice). Second, and once again, the labeling is odd, particularly in the case of GluR1, since it seems to surround the nucleus.           

Author Response

We appreciate and acknowledge a thorough review of our paper by both referees. Most comments made and the issues raised were fair and constructive. This manuscript is now revised in light of both reviewer’s recommendations; this referee’s concerns are addressed below on a point-to-point basis.

Reviewer 2:

Comments: This work examines the distribution of menin in the murine brain along development. It is exclusively descriptive and any possible functions of the protein in the context of brain development and function are simply inferred from its brain region or cellular localization, or its co-localization with other proteins. While the results presented by the authors could have some interest to the scientific community, there are many issues that need to be addressed. 

One of these is the language used. I strongly advise the authors to have the text proofed by a native English speaker or at least someone experienced. There are errors that should not have been found in a submitted manuscript, such as saying in the abstract that their study “is the first to provide an incompetent (the synonym of inadept) analysis[...]”.

Response: We appreciate referee’s recommendations, and this manuscript has now been thoroughly edited for the language, spellings and grammatic errors.

Comments: Introduction:

The introduction shows what is known in the field with relevance for the current study, but it is often written in an odd way. For example, stating the objectives as a long list of questions seems awkward.

Response: As per the referee’s suggestions, this section is now revised, and the question format is replaced with objectives that are now stated explicitly.

Methods:

The methods are generally written with enough detail, but some parts are not clear, as listed:

The mouse strain should be spelled C57BL/6;

Response: as per the referee’s suggestions, the mouse strain is now duly spelled out.

Neurons are dissociated, not detached;

Response: The term corrected.

 “plated at low to medium density” is vague and actual density /mm2 or cm2 should be given

Response: Changed as per the suggestions.

“Negative controls were performed to test the specificity of the antibodies”. Which controls specifically? NO detail is given and no results either, which seem to be part of a previous publication. Therefore, this data should either be shown or cited.

Response: The previous study is now cited, and additional knockdown data has been added as further verification.

Using fluorescence intensity as AU units to quantify menin density without specifying the range of AU is not useful

Response: The range is now highlighted in the methodology section

Results:

Section 3.1 is not a results section and should just be omitted, since the antibodies were tested in a previous study.

Changed accordingly

Comments: One major problem I see with the data is that the immunohistochemistry data is not very convincing, since the antibodies seem to label almost everything and everywhere. A proper control using neurons from KO mice would be very welcome to demonstrate staining specificity, especially since both antibodies label more than 1 band in western blots, as shown previously by the authors. The authors do mention negative controls, but these are not shown.

Response: The referee has raised an important issue and the requested data has now been added accordingly. Specifically, figure 1 has now been added to show hippocampus specific MEN1 deletion in homozygous Men1 floxed mice using Cre virus. The figure demonstrates the specificity of the antibody staining as the expression of menin protein was significantly downregulated in hippocampus versus in the cre negative controls (shown using confocal microscopy images).

Comments: The N fragment of menin is found only in the nucleus, whereas the C fragment seems to be everywhere, including the nucleus (Fig 1B). However, the authors’ description is misleading since it makes no mention of menin C also being nuclear. Also, this data is similar to the one in their previous publication (Angela et al, 2017) and could probably be omitted altogether.

Response: We would like to argue that N menin and C- menin have not been shown in a whole brain before – only in cultured neurons. Therefore, the data is shown here in Figure 2 to demonstrate for the first time both fragments being present in the intact brain. Moreover, C-menin is also localized to nuclear region and this has now been clarified in the manuscript at multiple places.

Comments: The authors also claim that the expression of menin is highest in the hippocampus, cortex, cerebellum and thalamus. However, no data is presented for any other brain regions. To make this claim, other brain regions should have been quantified and displayed, and a full brain section (coronal or sagittal) reconstructed from a mosaic of confocal images is lacking for the reader to appreciate the pattern of menin distribution in the brain. By only displaying magnified views of selected regions could show data bias.

Response: We appreciate this insightful comment, and a Figure 5 has now been added with the negative data to depict the areas where menin expression is the highest as compared to those areas where it’s significantly low or absent altogether.

Comments: Regarding the figure legends, confocal microscopy is referred to as “super high resolution confocal” microscopy. This does not exist, and the scale bars show that “normal” confocal microscopy was used. Super resolution microscopy allows resolutions of 200 nm or under, which cannot be attained with confocal microscopy (not at least without special sample preparation, as in expansion microscopy).

Response: The terminology has been corrected throughout the text

Comments: Section 3.3 says that menin C fragment is specifically expressed in neurons, but section 3.4 says that it is specifically localized to astrocytes. It would appear that the protein is localized everywhere and this could be simplified by just saying that it is present in both neurons and astrocytes.

Response: A summary statement has now been added at the end of the two sections to indicate that it’s located in both neuron and non-neuronal cells.

Comments: Data regarding the localization of menin expression during embryonic and postnatal development would also greatly benefit from a whole brain section, to show possible localized patterns in specific brain regions.

Response: The specific breakdown of menin in different regions of the brain during embryonic period to postnatal period is now shown in details in the supplementary data (see fig S1-S5).

Comments: The co-localization data with syt and PSD95 is extremely unconvincing. What do the authors mean by a “synaptic ROI”? Surely the authors don’t think that the area surrounded by a circle in fig 5 corresponds to synapses? The scale bar reads 20 um, which is the size of whole neurons, not synapses. The labeling appears to be intracellular, surrounding nuclei in what could be interneurons, but it’s unclear from what hippocampal region the images are from. Also, the images in the red channel and green channels seem to have greatly different pixel sizes, so even if the labeling were of synapses, co-localization would be hard to interpret in such conditions. The authors should have used higher magnification images and performed colocalization analysis at the single puncta level. The authors should specify which syt isoform is being labeled, since some are postsynaptic and be aware that no syt is a de facto presynaptic marker. Bassoon or other presynaptic active zone protein would have been a better choice.

Response: The hippocampal slice was used to quantify the colocalization of menin with both pre- and post-synaptic proteins.

The circles have been adjusted to show regions of boutons localization marking the putative synapses. A clearer image with higher magnification (63X) has been added to depict the expression of PSD-95 and its colocalization with menin protein.

In this study, we used SYT-1 which is a presynaptic calcium sensor protein indispensable for synaptic vesicle exocytosis mediating neurotransmitter release in hippocampal neurons[1,2]. We agree that demonstrating the localization of Bassoon would have been another interesting potential candidate but this protein has either been shown to be located at the photoreceptor Ribbon synapses in Retina or GABA-ergic or GluR1 neuronal synapses in the hippocampus. In this study, our focus was primarily on the nAChRs, and at these synapses SYT-1 remains the most desired candidate.

Comments: The data on excitatory vs inhibitory localization of menin in unconvincing. First, the authors fail to mention where in the brin they are imaging (it’s only referred to as synaptic ROI of an adult brain slice). Second, and once again, the labeling is odd, particularly in the case of GluR1, since it seems to surround the nucleus.      

Response: We would like to argue that the hippocampal slices were imaged to quantify the expression at excitatory versus inhibitory neurons. The GluR1 labelling is consistent with previous studies which illustrated its expression in hippocampal neurons[3]. A higher magnification image has now been provided in the figure to represent the puncta colocalization of GluR1 with menin.

  1. Jahn, R.a.F.D. Molecular machines governing exocytosis of synaptic vesicles. Nature (London) 2012, 490, 201--207.
  2. Südhof, T.C.a.R.J. Synaptotagmins: C2-domain proteins that regulate membrane traffic. Neuron (Cambridge, Mass.) 1996, 17, 379--388.
  3. Hagihara, H.a.O.K.a.T.K.a.M.T. Expression of the AMPA Receptor Subunits GluR1 and GluR2 is Associated with Granule Cell Maturation in the Dentate Gyrus. Frontiers in neuroscience 2011, 5, 100--100.

Round 2

Reviewer 2 Report

The authors have addressed all my concerns and, beyond some minor typos still remaining, I'm fine with the new version and their answers